# Soil mixing with organic matter amendment improves Albic soil physicochemical properties and crop yield in Heilongjiang province, China

Qingying Meng [1,2], Hongtao Zou[1]*, Chunfeng Zhang[2], Baoguo Zhu[2], Nannan Wang[2], Xiaohe Yang[1,2], Zhijia Gai[2], Yanyu Han[1]

1 Department of Land and Environment, Shenyang Agricultural University, Shengyang, Liaoning, China,
2 Jiamusi Branch, Heilongjiang Academy of Agricultural Sciences, Jiamusi, Heilongjiang, China

* zhtsynd@163.com

## Abstract

Crop productivity in Albic soil is poor, owing to poor soil physicochemical properties. Mixing of Aw layers, representing Albic soil, with B layers, could improve the physicochemical properties of Albic soil, which is characterized by poor humus on the topsoil and high penetration resistance. The objective of the present study conducted in 2015–2016 in an Albic soil region in Heilongjiang province, China, was to explore the effects of different soil mixing strategies on the physicochemical properties of Albic soil and crop yield. There were four soil mixing treatments: conventional subsoiler (CS), three-stage subsoil mixing plough (TSMP), four-stage subsoil mixing plough (FSMP), and three-stage subsoil interval mixing plough (TSIMP). Our results demonstrated that the Aw layer bulk density of Albic soil under TSMP, FSMP, and TSIMP decreased significantly compared to that under CS. In addition, the total porosity of the soil under these treatments increased significantly in 2 years. Compared to the water holding capacity under the CS treatment, other treatments increased significantly in the Aw layer. Furthermore, soil penetration resistance of the Aw layer decreased following Aw and B layer mixing. All three soil mixing treatments also increased soil aggregate stability and cation exchange capacity but reduced soil organic carbon content in the Aw layer. Soil mixing increased soybean and maize seed yield. Overall, Aw and B layer mixing improved Albic soil structure and physiochemical properties and increased crop yield; thus, this mixing is a feasible approach for Albic soil improvement, with optimal improvements observed under the FMSP strategy, which also added organic substances to the Aw layer.

## Introduction

Albic soil has a typical white-gray Albic layer (Aw layer). In China, Albic soil is mainly distributed in the Heilongjiang and Jilin provinces. In the Heilongjiang province, the total area under

**Data Availability Statement:** All relevant data are within the paper and its Supporting Information files.

**Funding:** The work supported by the Heilongjiang Academy of Agricultural Sciences "Agricultural science and technology innovation leap project", special technology for sustainable utilization of black soil cultivated land resources (HNK2019CX13)".

**Competing interests:** The authors have declared that no competing interests exist.

Albic soil is 3.31 million ha [1]. Albic soil in northeast China is similar to soils described as "Lessivage" (France), "Pseudo-gley" (Germany), "pseudopodzolic" (Russia), and certain clay pan soils (USA) [2]. In earlier studies, Albic soil of Northeast China has been considered podzolic or solodic soil [2, 3]. Zheng (1963) pointed out that Albic soil originates from periodic waterlogging, and its formation in Northeast China is driven by eluviation [2]. In Chinese soil taxonomy, Albic soil is classified under the Alfisol class, the cold Alfisol subclass, and the Albic cold Luvisol class [4]. More than 30 countries have soils with an Albic layer or Albic material across the globe [5, 6].

Crop production in the Albic soil region is poor. For example, soybean seed yield has been reported to be 20% lower when grown under Albic soil than under black soil [7, 8]. Two key factors may explain the poor crop yields observed under Albic soils. First, there is a relatively thin humic soil layer (the Ap layer) with a low thickness of about 20 cm near the surface in Albic soils. Secondly, plant roots generally cannot penetrate the underlying Aw layer [9] because of its high penetration resistance, which ranges from 2 MPa to more than 2.5 MPa [10]. In addition, the movement of water and air molecules in the topsoil and the sublayer is impaired, which further reduces root growth in the soil layer. The water regime in the Albic layer soil is poor, with only 5.39% available moisture [11], and soil permeability has been reported to be $6.08 \times 10^{-5}$ cm s$^{-1}$ at saturation [10]. The soil layer thickness conducive for plant growth in dry farmland Albic soil in the Heilongjiang province has been reported to be 20 cm [1].

Over the past three decades, researchers have attempted to improve Albic soil, and in turn, increase crop yield, primarily by increasing organic matter concentrations in the Ap layer and the depth of available topsoil. For example, researchers have amended Albic soil with straw [12, 13], biochar [14, 15], and organic fertilizers [16], including green manure and chemical fertilizers [17], to increase organic matter concentrations and the nutrients available to plants. In addition, subsoiling and super subsoiling have been applied extensively in agricultural production in Albic soil regions [18, 19], and they could increase soil nutrients or crop yield [15]; however, addressing the poor humus levels in the topsoil and high penetration resistance of the Aw layer that represents Albic soil remains a challenge.

Zhao (1989) conducted field tests in wheat plots under Aw and illuvial (B) layers of Albic soil mixed at different ratios. According to the results, the optimal Aw:B ratios were 1:1 and 0.5:1, which increased wheat yield by 4.5% [20]. Consequently, a three-stage subsoil mixing plough (TSMP) was developed in the mid-1990s to improve Albic soil [21]. The TSMP can mix the Aw and B layers of Albic soil at a 1:1 ratio while leaving the Ap layer undisturbed [20–23]. In contrast to the typical subsoiling technique, subsoil hardness did not return to the initial level; therefore, Aw soil permeability increased 1.7–7.3-fold, and soybean yield increased by 11.4% [24]. A Four-stage subsoil mixing plough (FSMP), which applies similar principles to those of the TSMP for improving soil, was developed in 2002 [25, 26]; however, it could also apply wheat straw, maize straw, and fertilizer into the 20–40-cm soil layer [25]. The large-scale adoption of the TSMP and FSMP strategies has been limited because of their low operation efficiencies [24]. In addition, a three-stage subsoil interval mixing plough (TSIMP) was developed in 2010. It represented an increase in working width from 46 cm (TSMP) to 92 cm, and it improved soil improvement efficiency, increasing soybean yield by 4.5% when compared with CS [24]. Although such developments have provided mechanical and theoretical frameworks for the improvement of Albic soil, there is a lack of comparative studies based on the physicochemical properties of Albic soils managed using different Subsoil Mixing ploughs. Consequently, the objective of the present study was to compare different methods of mixing the Aw and B layers by investigating their effects on the physicochemical properties of Albic soil, in addition to soybean and maize yield.

## Materials and methods

### Study site and soil description

The experimental site is located in State Farm 853 in Heilongjiang Province, China (42˚30′N, 136˚35′E). The farm is in a complex transition zone between the Three-River-Plain and Wanda Mountain. Soil types at the site include Albic soil, meadow soil, bog soil, dark-brown soil, and black soil. The site experiences the monsoon climate of a sub frigid zone. Winters are cold and dry, while summers are hot and rainy. The average annual temperature is 2.4˚C and the accumulated temperature $> 10$˚C is 2442.8˚C. The mean annual precipitation is 500–600 mm. Before implementing the treatments, soil organic carbon (SOC), C/N ratio, and pH data in the 0–20 cm soil layer were collected in Oct 2014 and were 24.46 g kg$^{-1}$, 13.13 and 6.3, respectively.

The soil layers of Albic soil are clearly distinct in the solum. The Albic soil in the State Farm 853 in the Heilongjiang province exhibited the typical Ap-Aw-B-C profile (Fig 1) From the surface to a depth of 15–20 cm, the soil exhibited a silty or clay loam Ap layer, with a blocky structure and a black color. The Ap layer was rich in organic matter and conducive for plant root system growth. From the 15–20 cm to the 18–22 cm depth, the silty Aw layer exhibited a lamellar structure and a white-gray color. The Aw layer represents Albic soil. The Albic soil is dense, impermeable to water, and inhibits crop growth. The B horizon is below the Aw layer at a depth of 40–55 cm and is composed mainly of heavy clay soil. The third layer (C) is the parent soil, which consists of yellow clay, at a depth of 5–11 m [10, 26].

### Experimental design

The experimental field was established in Autumn 2014. Four tillage systems were used for the experiments: conventional subsoiler (CS) treatment, achieved using a conventional subsoiler that which served as the control treatment and three experimental tillage treatments, including three-stage subsoil mixing plough (TSMP), four-stage subsoil mixing plough (FSMP), and three-stage subsoil interval mixing plough (TSIMP). No tillage was performed in 2015–2016 years of this experiment. TSMP can mix Aw and B soil layers at an approximate ratio of 1: 1 below the surface leaving the Ap layer undisturbed; FSMP is similar to TSMP, although with an additional 1.50 kg m$^{-2}$ of maize straw returned to the 20–40 cm soil layer. In addition, TSIMP is similar to TSMP; however, in TSIMP, soil is mixed across a 92-cm-wide site with alternating mixed and non-mixed areas, with a non-mixed 62-cm buffer between the intervals. Fig 2 is a schematic of the four treatments, which were assigned based on a completely random design with three replicates. Each plot had an area of 65 m$^2$ (6.5 m ×10 m).

Soybean was planted in May 2015 and harvested in September 2015, while maize was planted in May 2016 and harvested in September 2016. Soybean cultivar Kenfeng 16 was planted at a rate of 280,000 seeds ha$^{-1}$ using a planter. Maize cultivar Jidan 27 was planted at a rate of 500,000 seeds ha$^{-1}$ using a planter. For soybean, 50 kg N ha$^{-1}$, 100 kg P ha$^{-1}$, and 30 kg K ha$^{-1}$ were applied as base fertilizers, and the base fertilizers were applied during sowing. For maize, 90 kg N ha$^{-1}$, 100 kg P ha$^{-1}$, and 60 kg K ha$^{-1}$ were applied as base fertilizers and an additional 80 kg N ha$^{-1}$ applied as top-dressing.

### Soil sampling and analysis

Ap, Aw, and B soil layer samples were collected using S-type methods at five points and 0–-20-cm, 20–40-cm, and 40–60-cm layers, respectively, from each plot immediately after soybean and maize harvest in 2015 and 2016, respectively. Five soil cores were collected using a hand auger in each plot and then homogenized. The soil samples were air-dried at room

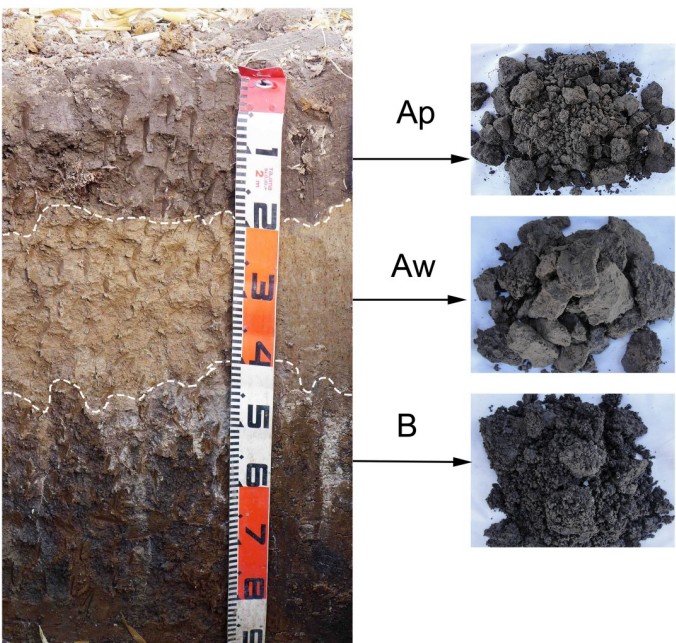

**Fig 1. Typical Albic soil at State Farm 853, Heilongjiang China.** Ap: topsoil (0–20 cm), Aw: Albic soil layer (20–40 cm); B: illuvial soil layer (40–60 cm).

temperature and crushed through a 2-mm sieve for soil cation exchange capacity (CEC) and SOC analyses. Undisturbed soil samples were collected from the Ap and Aw layers using a wide-blade knife at five points and placed in sampling boxes, and these samples were mixed for later soil aggregate measurements. Undisturbed 100 cm$^3$ soil cores were collected from Ap, Aw, and B layers in each plot. The soil samples were used for soil bulk density and soil three-phase ratio analyses.

Soil bulk density was determined using routine methods [27]. A soil three-phase volumen-ometer (DIK-1120; Daiki Rika Kogyo Co. Ltd., Japan) was used to determine the three-phase ratio in the soils. The soil samples were collected in each year, and soil penetration resistance was determined using a cone penetrometer (DIK-5521; Daiki Rika Kogyo Co. Ltd.), which was driven vertically into the soil by hand to a depth of about 60 cm, with soil penetration resis-tance recorded continuously. The water-stable aggregates were measured using the wet aggre-gate by a sieving apparatus (DIK-2000, Daiki Rika Kogyo Co. Ltd.). The soil samples were separated into four classes, namely >2 mm, 0.25–2 mm, 0.053–0.25 mm, and <0.053 mm clas-ses. All the isolated fractions were weighed and used to calculate soil mean weight diameter (MWD) according to Cabardella and Elliott (1993) [28]. CEC was measured by leaching the

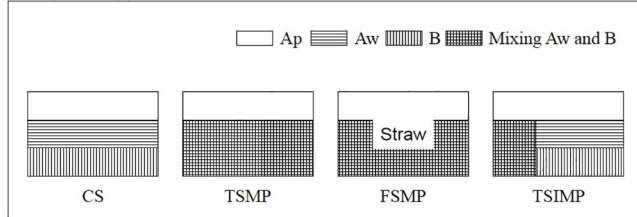

**Fig 2. Schematic diagram of the four treatments.**

dried soil samples with 1 M ammonium acetate, and analyzing the leachate using methods described by Lu [29]. SOC was determined by dichromate oxidation [29].

## Grain yield and yield components

The yield components evaluated were pod number per plant, grain number per plant, and 100-grain weight. Two central rows of soybean and maize in each plot were harvested for the determination of grain yield at 13% grain moisture.

## Statistics

Data were analyzed by one-way Analysis of Variance and the Least Significant Difference test was used to differentiate the means. The statistical analyses were performed using IBM SPSS Statistics 19.0 (IBM Corp., Armonk, NY, USA). All the figures were plotted by Origin 9.0 (OriginLab Corp., Northampton, MA, USA).

# Results

## Soil bulk density and pore characteristics

As shown in Table 1, mixing the Aw and B layers decreased the soil bulk density of the Aw layer significantly ($P < 0.05$). The bulk density of the Aw layer was CS > TSIMP > TSMP > FSMP in 2015, and was in the order of CS > TSIMP > FMSP > TSMP in 2016. Mixing the Aw and B layers had no significant effect on the soil bulk density of Ap and B layers in 2 years.

The Aw layer of the Albic soil had a low total porosity (Table 2). The total porosity of the Aw layer in the TSMP, FMSP, and TSIMP treatments had increased compared to CS, respectively, in 2015. Similar results were observed in 2016. The capillary porosity of the Aw layer in the TSMP, FMSP, and TSIMP treatments was significantly higher than in the CS for 2 years. The air-filled porosity of the Aw layer in the TSMP and FSMP treatments were significantly higher than in CS in 2 years of the experiment. Mixing the Aw and B layers had little influence on Ap layer and B layer.

The variation of total porosity with the soil layer is opposite to that of soil bulk density. For instance, the bulk density of CS was Aw > B > Ap layer in 2015 and 2016, but the total porosity was Ap > B > Aw layer.

## Soil water holding capacity

The Albic soil water holding capacities wee low especially in the Aw layer, as shown in Fig 3. The Albic soil water holding capacities of the Aw layer in TSMP and FSMP were significantly increased compared to CS in 2 years. The effect on the water holding capacity was greater for

**Table 1. Effects of soil mixing on the bulk density of Albic soil.**

| Treatment | 2015 | | | 2016 | | |
|---|---|---|---|---|---|---|
| | Ap (g cm⁻³) | Aw (g cm⁻³) | B (g cm⁻³) | Ap (g cm⁻³) | Aw (g cm⁻³) | B (g cm⁻³) |
| CS | 1.26 ± 0.01 a | 1.50 ± 0.01 a | 1.37 ± 0.03 a | 1.23 ± 0.01 a | 1.48 ± 0.01 a | 1.35 ± 0.01 a |
| TSMP | 1.23 ± 0.04 a | 1.37 ± 0.03 b | 1.37 ± 0.03 a | 1.20 ± 0.02 a | 1.34 ± 0.02 c | 1.34 ± 0.01 a |
| FSMP | 1.25 ± 0.01 a | 1.35 ± 0.02 b | 1.34 ± 0.02 a | 1.22 ± 0.04 a | 1.35 ± 0.02 c | 1.36 ± 0.02 a |
| TSIMP | 1.24 ± 0.01 a | 1.41 ± 0.03 b | 1.39 ± 0.01 a | 1.23 ± 0.04 a | 1.42 ± 0.02 b | 1.35 ± 0.02 a |

Different lowercase letters indicate significant differences between samples ($P < 0.05$). Values are means ± standard errors (n = 3).

**Table 2. Effects of soil mixing on the soil characteristics of Albic soil.**

| Treatment | | 2015 | | | 2016 | | |
|---|---|---|---|---|---|---|---|
| | | Ap | Aw | B | Ap | Aw | B |
| Soil porosity (%) | CS | 56.17 ± 1.04 a | 39.37 ± 0.27 c | 48.68 ± 0.48 a | 56.70 ± 0.77 a | 41.16 ± 0.61 c | 47.54 ± 0.69 b |
| | TSMP | 54.19 ± 0.36 a | 46.60 ± 0.52 a | 47.03 ± 0.37 a | 57.27 ± 0.64 a | 49.09 ± 0.67 b | 48.96 ± 0.30 ab |
| | FSMP | 54.95 ± 0.85 a | 47.60 ± 0.76 a | 47.55 ± 0.78 a | 56.31 ± 0.83 a | 51.32 ± 0.67 a | 49.27 ± 0.31 a |
| | TSIMP | 54.35 ± 0.89 a | 44.73 ± 0.56 b | 48.90 ± 0.81 a | 57.38 ± 0.49 a | 48.87 ± 0.71 b | 49.68 ± 0.41 a |
| Capillary porosity (%) | CS | 41.72 ± 0.95 a | 33.44 ± 0.65 b | 38.11 ± 0.74 a | 41.38 ± 1.03 a | 34.67 ± 1.14 b | 36.12 ± 0.26 a |
| | TSMP | 40.08 ± 0.08 a | 37.68 ± 0.69 a | 38.62 ± 0.91 a | 40.58 ± 0.43 a | 39.29 ± 0.98 a | 36.32 ± 0.58 a |
| | FSMP | 40.33 ± 0.47 a | 38.29 ± 0.45 a | 37.10 ± 0.44 a | 40.50 ± 0.86 a | 41.89 ± 1.22 a | 35.71 ± 0.72 a |
| | TSIMP | 40.59 ± 0.82 a | 37.21 ± 1.10 a | 37.71 ± 0.93 a | 40.67 ± 1.24 a | 40.43 ± 0.81 a | 37.82 ± 1.11 a |
| Air-filled porosity (%) | CS | 14.46 ± 0.21 a | 5.93 ± 0.75 b | 10.57 ± 0.54 a | 15.32 ± 0.41 a | 6.49 ± 0.57 b | 11.42 ± 0.67 b |
| | TSMP | 14.10 ± 0.38 a | 8.92 ± 0.19 a | 8.41 ± 0.54 a | 16.69 ± 0.22 a | 9.80 ± 0.39 a | 12.64 ± 0.47 ab |
| | FSMP | 14.62 ± 0.75 a | 9.30 ± 0.35 a | 10.45 ± 0.46 a | 15.81 ± 0.50 a | 9.43 ± 0.56 a | 13.55 ± 0.45 a |
| | TSIMP | 13.76 ± 0.26 a | 7.53 ± 0.73 ab | 11.19 ± 1.55 a | 16.71 ± 0.78 a | 8.44 ± 0.49 a | 11.86 ± 0.74 ab |

Different lowercase letters indicate significant differences between samples ($P < 0.05$). Values are means ± standard errors (n = 3).

FSMP than TSMP in 2 years. Mixing of the Aw and B layers had no significant effect on the water holding capacity of Ap and B layers in 2 years.

## Soil penetration resistance

Albic soil penetration resistance values over a two-year period are presented in Fig 4. The penetration resistance of the Ap layer of the Albic soil was low, but it increased in the Aw layer and declined again in the B layer. The soil penetration resistance in the CS treatment was 2.03 MPa at 25 cm, which was significantly higher than the soil penetration resistance values in the

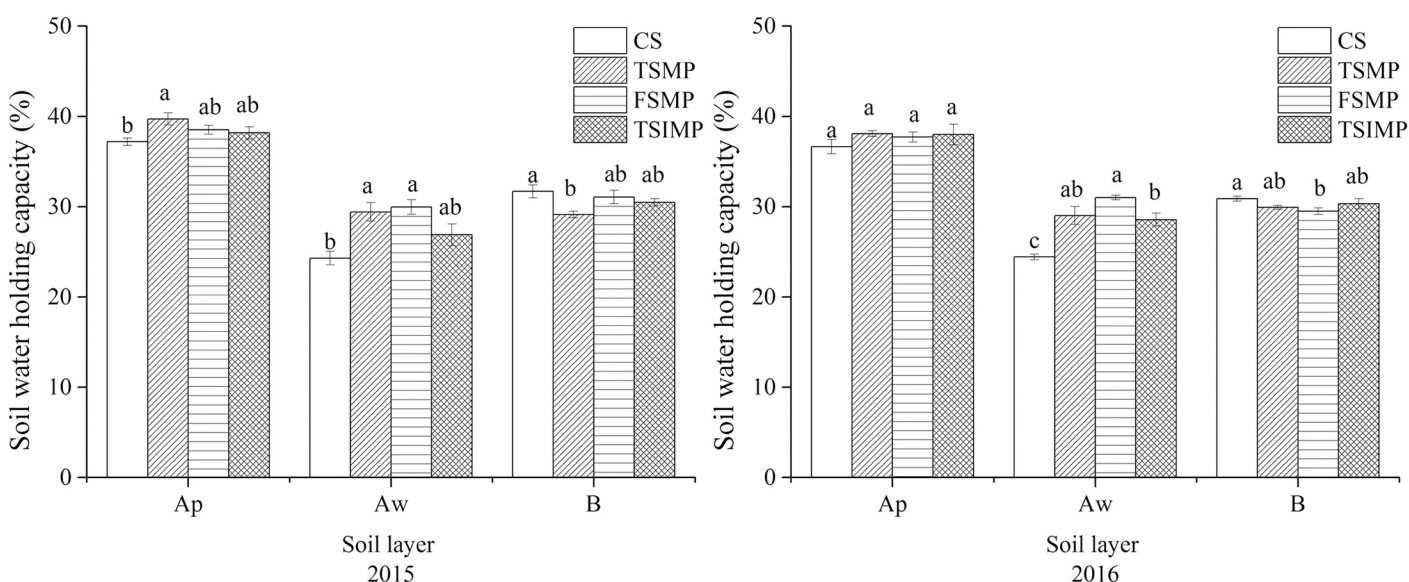

**Fig 3. Effects of soil mixing on the soil water holding capacity of Albic soil.** Different lowercase letters indicate significant differences between samples ($P < 0.05$). Bars represent standard errors (n = 3).

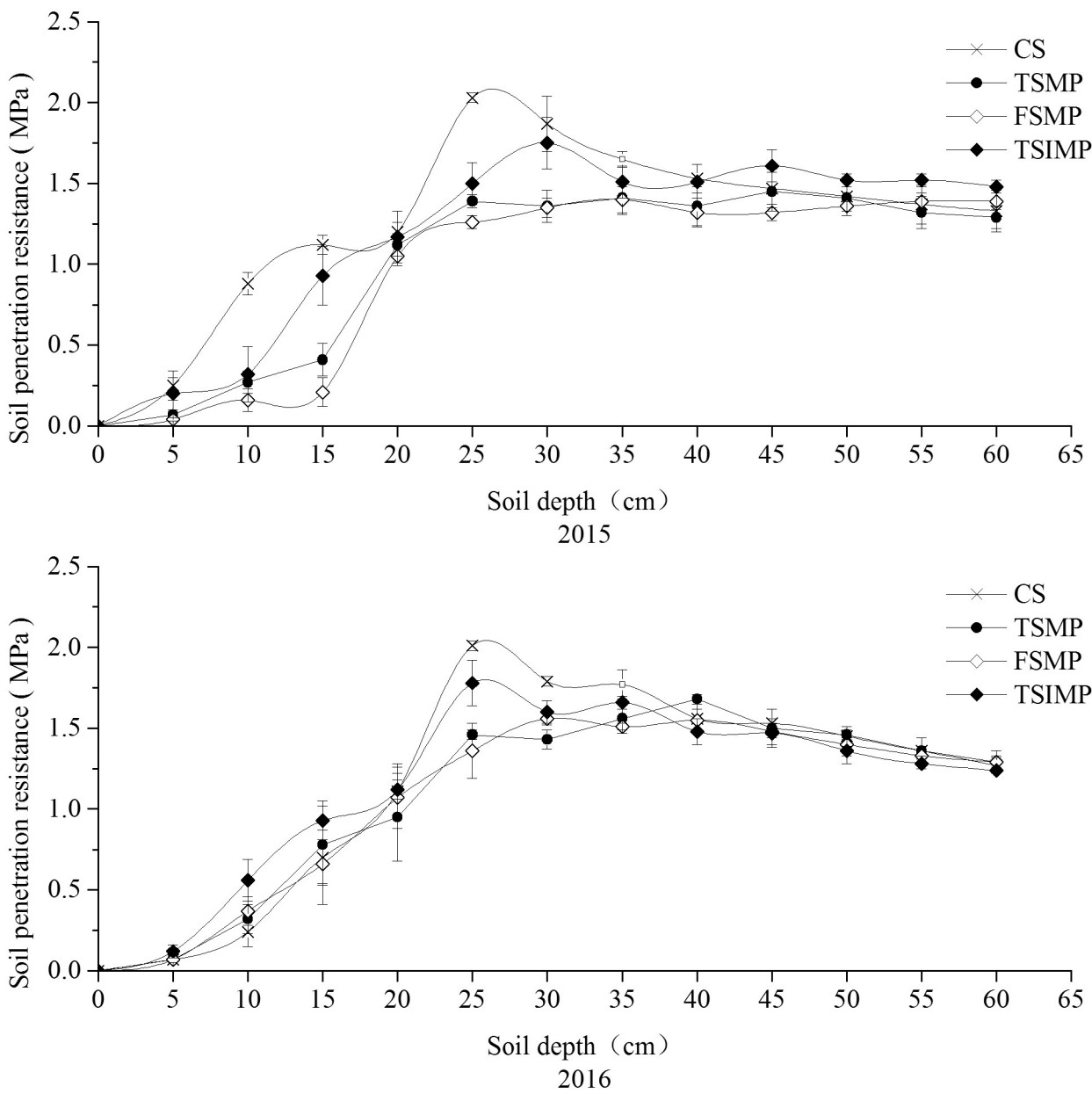

**Fig 4. Effects of soil mixing on the soil penetration resistance of Albic soil.** Bars represent standard errors (n = 3).

TSMP, FMSP, and TSIMP treatments at 25 cm ($P < 0.05$), including 1.39 MPa, 1.26 MPa, and 1.50 MPa, respectively. The soil penetration resistance of the Aw layer in the CS treatment at a 25-cm depth was 2.01 MPa. The penetration resistance values in the TSMP, FMSP, and TSIMP treatments, were much lower than those in the CS treatment at a 25-cm depth and were 1.46 MPa, 1.36 MPa, and 1.78 MPa, respectively.

## Water-stable aggregate distribution and stability

The separation of the soil samples collected in September 2015 and 2016 into size classes based on water-stable aggregates revealed similar proportions of aggregates of different

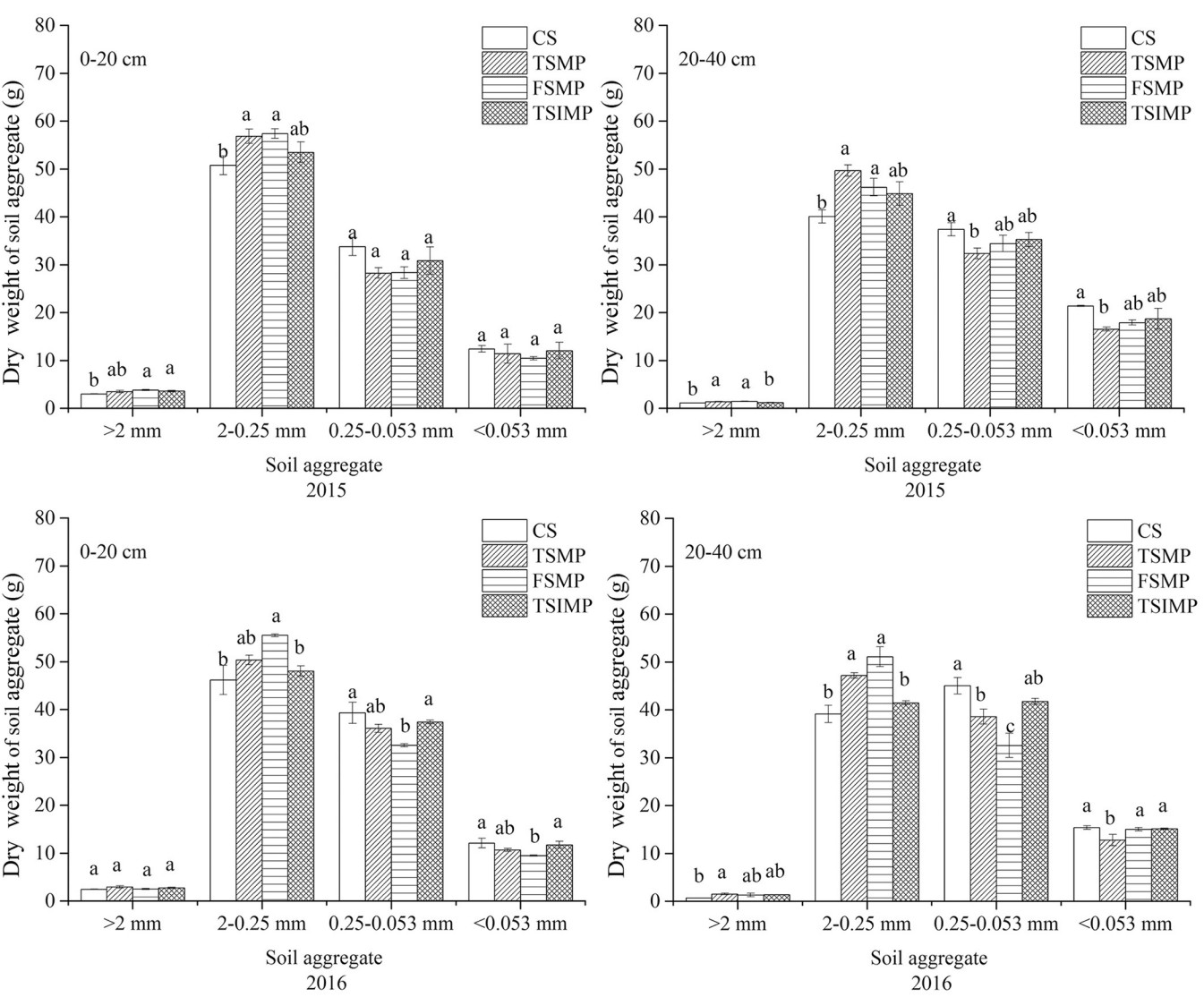

**Fig 5. Effects of soil mixing on the water-stable aggregate of Albic soil.** Different lowercase letters indicate significant differences between samples ($P < 0.05$). Bars represent standard errors (n = 3).

sizes between the Ap and the Aw layer (Fig 5). Small macroaggregates (2–0.25 mm) accounted for the largest fraction in all treatments (50.07–70.77%), and the large macroaggregates (>2 mm) accounted for the lowest fraction (0.92–4.30%). Microaggregates (0.25–0.053 mm) and free silt and clay (<0.053 mm) accounted for 16.92–34.52% and 8.23–20.81% of the fractions, respectively. The >0.25 mm aggregate and MWD are commonly used to evaluate the soil aggregate stability [30]. Compared with CS, other treatments increased the relative proportions of the 2–0.25 mm aggregate and the MWD (Fig 6). The highest proportions of 2–0.25 mm aggregates were observed in the FMSP treatment in both the Ap and Aw layers. In addition, in both the Ap and Aw layers, the MWD of the CS treatment was lower than those of the other treatments. In 2015, the highest MWD in the Ap layer was observed under the FMSP treatment in the Ap layer, while, in 2016, the highest MWD in the Ap layer was observed in the TSMP treatment.

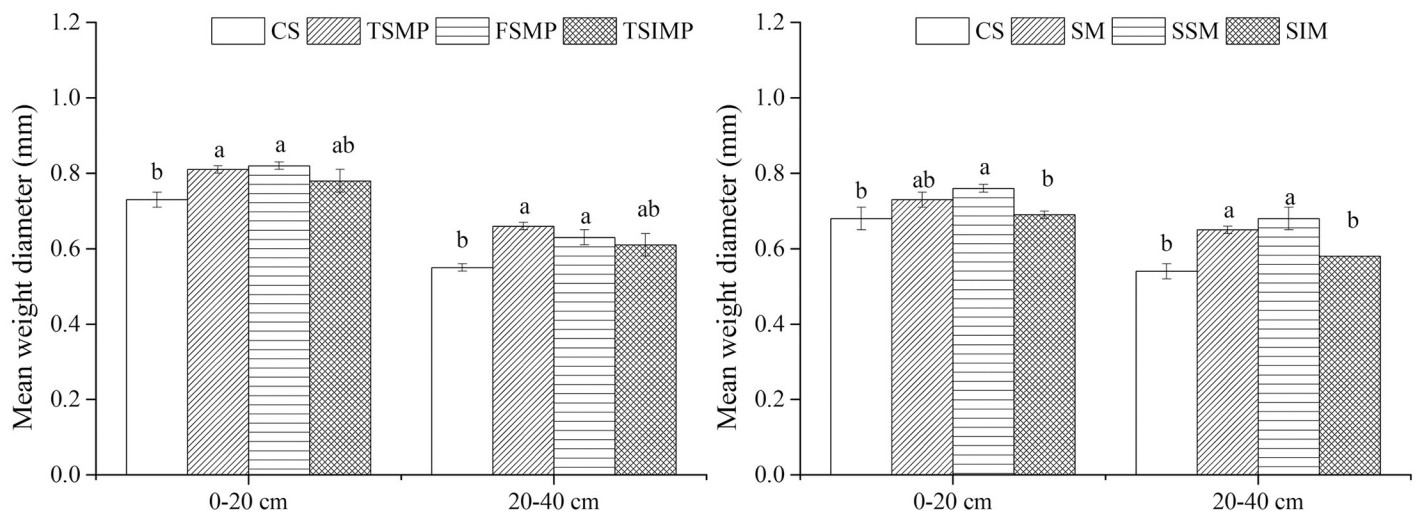

**Fig 6. Effects of soil mixing on the MWD of Albic soil.** Different lowercase letters indicate significant differences between samples ($P < 0.05$). Bars represent standard errors (n = 3).

## Soil CEC

Mixing of the Aw and B layer soil affected the soil CEC. Soil CEC of the Ap layer was about 20 cmol kg$^{-1}$ in the four treatments (Table 3). In 2015, compared to CS, soil CEC of the Aw layer in the TSMP, FMSP, and TSIMP treatments were significantly higher ($P < 0.05$) by 14.73%, 15.70%, and 10.85%, respectively. In 2016, TSMP, FMSP, and TSIMP increased by 15.36%, 16.26%, and 10.14%, respectively. In contrast, soil CEC in the B layer decreased across the treatments.

## Soil organic carbon

Table 4 illustrates the significant decreases in SOC concentrations with a decrease in soil depth, where SOC of the Ap layer > Aw layer > B layer. There were no significant differences in SOC concentrations in the Ap layer among treatments, while significant differences were observed between the Aw and B layers. The SOC concentrations in the Aw layer across treatments were ranked as follows: CS > FMSP> TSIMP > TSMP, in 2015, and CS > TSIMP> FMSP > TSMP, in 2016. Compared to CS, the SOC concentrations in the Aw layer decreased because of the mixing of the Aw and B layers in the FMSP, TSIMP, and TSMP treatments. The SOC concentrations in the B layer across the four treatments were in the order of FMSP >TSMP > TSIMP > CS in two years.

**Table 3. Effects of soil mixing on the cation exchange capacity of Albic soil.**

| Treatment | 2015 | | | 2016 | | |
|---|---|---|---|---|---|---|
| | Ap (cmol kg$^{-1}$) | Aw (cmol kg$^{-1}$) | B (cmol kg$^{-1}$) | Ap (cmol kg$^{-1}$) | Aw (cmol kg$^{-1}$) | B (cmol kg$^{-1}$) |
| CS | 20.46 ± 0.40 a | 15.48 ± 0.76 b | 27.22 ± 0.11 a | 20.49 ± 0.36 a | 15.49 ± 0.13 b | 26.24 ± 0.21 a |
| TSMP | 20.79 ± 0.27 a | 17.76 ± 0.28 a | 22.17 ± 0.37 b | 20.17 ± 0.32 a | 17.87 ± 0.32 a | 22.35 ± 0.66 c |
| FSMP | 20.30 ± 0.32 a | 17.91 ± 0.26 a | 22.62 ± 0.31 b | 20.07 ± 0.30 a | 18.00 ± 0.20 a | 24.03 ± 0.51 b |
| TSIMP | 20.70 ± 0.81 a | 17.16 ± 0.28 a | 23.06 ± 0.51 b | 20.21 ± 0.54 a | 17.06 ± 0.43 a | 25.28 ± 0.23 ab |

Different lowercase letters indicate significant differences between samples (p <0.05). Values are means ± standard errors (n = 3).

**Table 4. Effects of soil mixing on the organic carbon of Albic soil.**

| Treatment | 2015 | | | 2016 | | |
|---|---|---|---|---|---|---|
| | Ap (g kg⁻¹) | Aw (g kg⁻¹) | B (g kg⁻¹) | Ap (g kg⁻¹) | Aw (g kg⁻¹) | B (g kg⁻¹) |
| CS | 23.04 ± 0.64 a | 15.37 ± 0.28 a | 8.87 ± 0.12 b | 23.50 ± 1.10 a | 14.53 ± 0.44 a | 7.62 ± 0.48 b |
| TSMP | 23.23 ± 0.72 a | 13.69 ± 0.43 b | 9.76 ± 0.10 a | 23.12 ± 1.19 a | 12.34 ± 0.70 b | 9.19 ± 0.47 ab |
| FSMP | 23.34 ± 0.64 a | 14.73 ± 0.48 ab | 9.84 ± 0.25 a | 22.83 ± 0.66 a | 13.59 ± 0.80 ab | 9.99 ± 0.63 a |
| TSIMP | 23.49 ± 0.05 a | 14.06 ± 0.38 ab | 9.64 ± 0.14 a | 22.32 ± 1.23 a | 13.87 ± 0.31 ab | 8.29 ± 0.90 ab |

Different lowercase letters indicate significant differences between **samples** (p <0.05). Values are means ± standard errors (n = 3).

## Grain yield

Grain yield data are summarized in Table 5. In 2015 and 2016, grain productivity was greater in the TSMP, FMSP, and TSIMP treatments than in the CS. Soybean seed yields in the TSMP, FMSP, and TSIMP plots in 2015 were 5.31%, 7.43%, and 3.85% higher than the yields in the CS plot, respectively. The number of pods per plant of soybean in the FSMP treatment was higher than that in other treatments. ($P > 0.05$); however, the number of grains per pod was not significantly affected by mixing of the Aw and B layers of Albic soil ($P < 0.05$). The 100-grain weight was TSMP > FSMP > TSIMP > CS.

In 2016, the maize seed yields in the TSMP, FMSP, and TSIMP plots increased by 5.05%, 7.73%, and 2.19%, respectively. The number of grains per ear and 100-grain weight in the FSMP, TSMP, and TSIMP treatments were significantly higher than in CS.

## Discussion

The Ap layer and B layer of Albic soil have the same particle distribution. However, the Ap layer has low bulk density (1.06 g cm⁻³) [20], One possible reason is that the Ap layer has more organic matter; the Ap layer soil has better aggregate structures [31]. Research has shown that when 1% organic matter is mixed into soil, the bulk density decreases from 1.45 to 1.60 g cm⁻³ [32]. According to Araya (1991), the structure of the Albic soil layer, where clay fills the pore spaces of frame structures produced using silt particles, leads to a high bulk density (1.53 g cm⁻³) in the Aw layer of Albic soil, in addition to a low total porosity of 39.20% in the Aw layer of Albic soil. High bulk density and imbalanced pore distribution are the main factors contributing to the low productivity of Albic soil [11, 31]. To change the poor physical properties of the Aw layer, organic materials could be applied [12, 13], but it is difficult to get large

**Table 5. Effects of soil mixing on the crop yield of Albic soil.**

| Treatment | | No. of pods per plant | No. of grains per pod | 100-grain weight (g) | Grain yield (kg ha⁻¹) |
|---|---|---|---|---|---|
| 2015 Soybean | CS | 21.95 ± 0.13 b | 2.00 ± 0.04 a | 20.04 ± 0.41 b | 1842.10 ± 50.79 b |
| | TSMP | 21.75 ± 0.09 b | 2.05 ± 0.02 a | 21.57 ± 0.33 a | 2017.10 ± 56.07 a |
| | FSMP | 24.20 ± 0.58 a | 1.93 ± 0.08 a | 20.80 ± 0.42 ab | 2029.32 ± 33.06 a |
| | TSIMP | 22.71 ± 0.21 b | 2.00 ± 0.03 a | 20.60 ± 0.52 ab | 1955.37 ± 19.22 ab |
| | | No. of grains per ear | No. of ears per plant | 100-grain weight (g) | Grain yield (kg ha⁻¹) |
| 2016 Maize | CS | 508.89 ± 1.35 b | 1.19 ± 0.02 a | 34.40 ± 0.25 b | 8338.84 ± 64.20 b |
| | TSMP | 530.67 ± 5.59 a | 1.17 ± 0.03 a | 36.76 ± 0.76 a | 9151.74 ± 185.29 a |
| | FSMP | 528.00 ± 5.82 a | 1.20 ± 0.02 a | 36.41 ± 0.48 ab | 9310.74 ± 169.49 a |
| | TSIMP | 521.56 ± 6.91 ab | 1.17 ± 0.03 a | 35.70 ± 0.75 ab | 8733.38 ± 237.00 ab |

Different lowercase letters indicate significant differences between samples (p <0.05). Values are means ± standard errors (n = 3).

concentrations of organic materials. When the Aw and B layers were mixed, the soil configuration of the Aw layer was changed. The original Aw layer and B layer were changed to a mixed layer and B layer. The mixed layer had new soil configuration and decreased the bulk density and increased the total porosity and air-filled porosity and water holding capacity of the soil during the 2 years study.

According to the Japan Society of Soil Physics, a soil penetration resistance value of 2 MPa hinders root growth [33]. In the present study, the Aw layer of the Albic soil had a high soil penetration resistance of > 2 MPa. One possible reason for the high penetration resistance is that soil compaction occurred when small particles filled the voids within the frame structures of larger particles [33]. High soil penetration resistance has been previously reported to be associated with poor crop yields in an Albic soil region [7]. Compared to the penetration resistance in the CS treatment, the penetration resistance in the TSMP, FMSP, and TSIMP treatments was lower at the 20–40-cm layer in both 2015 and 2016.

The Albic layer soil has a high capacity for physical recovery, as observed under conventional subsoiling [7, 11]. Conventional subsoiling could loosen the Aw layer, but after 45 days and 11 times rainfall, the bulk density returns to normal [34]; however, mixing the Aw and B layers can hinder the physical recovery of the Albic soil layer because of the disturbed Albic layer structure. Most previous studies have focused on the Ap layer of Albic soil. If mixing the Aw and B layers could improve Albic soil physicochemical properties, the key limitations of Albic soil could be addressed.

The aggregate stability is affected by the parent material (clay sediments) of Albic soil. Albic soil has higher concentrations of microaggregates, and <0.25-mm microaggregate content has been reported to be 80–90% in the Ap layer, decreasing with a decrease in SOC [16, 35, 36]. In the present study, mixing Aw and B layer soils improved the proportion of the >0.25-mm aggregate portion and MWD. In addition, when compared to CS, other treatments increased the relative proportions of the 2–0.25-mm aggregates and the MWD significantly. The results demonstrate that mixing the Aw and B layer soils influences soil aggregation processes because mixing the two layers decreases the compactness of the soil structure.

Soil CEC is a key parameter that guides soil management activities under agricultural production. The soil CEC of the Aw layer was 60% lower than that of the B layer, suggesting that the fertilizer retention performance of the Aw soils was poor [19]. The CEC of the mixed soils increased following the mixing of the Aw and B layers. Because the B layer is a clay soil, it has a considerably large surface area for the adsorption of water and ions. Therefore, mixing of the two layers improved the nutrient retention capacity of the soil. Overall, mixing the two layers enhanced fertilizer retention and the buffer capacity of the Aw layer.

In a previous study, after 3 years of cultivation of an uncultivated Albic soil, the SOC of the Ap layer decreased from 35.50 g kg$^{-1}$ to 26.80 g kg$^{-1}$, and after 15 years of cultivation, the SOC decreased further to 19.83 g kg$^{-1}$, and after 30 years, the SOC content was maintained at 18.62 g kg$^{-1}$ [37]. SOC content is relatively high in natural Albic soil. Similarly, in another study, after cultivation and planting, SOC content decreased rapidly, slowly, and then gradually became stable [37]. The results indicated that the SOC of the Ap layer was not low, although the SOC content declined rapidly in the Aw layer under cultivation and crop production. In the present study, mixing the Aw and B layer soils improved the physical properties of the Aw layer. In addition, the soil mixing treatments altered SOC levels in the vertical soil layers. Compared to the SOC content in CS, the SOC content in the Aw layer in the FMSP, TSIMP, and TSMP treatments decreased because of the mixing of Aw and B layers. Consequently, soil mixing increased the uniformity of SOC distribution among the layers and increased SOC in the 0–60-cm soil layers. However, when mixing the Aw and B layers in an effort to improve Albic

soil, exogenous organic carbon should also be amended, for example, in the form of straw, to increase SOC contents in the new subsoil layer.

Grain yield was higher following the mixing of the Aw and B layers in the two-year study period than in the CS treatment. Crop straw, which could increase organic matter content [38] and aggregate stability [39, 40], should be incorporated into the subsoil during Aw and B layer mixing. The results of the present study indicate that the positive effects of mixing the Aw and B layer soils on improved crop yield in Albic soil using different subsoil mixing ploughs [21, 24, 25] (i.e., in the TSMP, FMSP, and TSIMP treatments) could be sustained for at least two years. Although using a subsoiler method (i.e., the CS treatment) can also facilitate Albic soil improvement [16, 19], the positive effects last a relatively short period and the mixing has to be undertaken annually. Nevertheless, considering that the effects of altered soil aggregate distribution following soil mixing persists for two years or more, planting costs could be reduced drastically. Therefore, according to the results of the present study, mixing of the Aw and B layer soils could enhance crop productivity under Albic soil because altered soil aggregate distribution and the associated positive effects can persist for more than 2 years. At present, we do not know the extent to which the positive effects of mixing the Aw and B layer soils can be attributed to changes in microbial community composition and function resulting from mixing. Future research will examine the impacts of mixing on the soil microbial community and the role of the soil microbial community in improving Albic soils.

## Conclusions

1. Mixing the Aw and B layers could reduce soil bulk density and increase soil total porosity in the Aw layer, which would improve soil water holding capacity. In addition, soil penetration resistance declines with a decline in the solid phase proportions in the Aw layer.

2. Mixing of the Aw and B layers alters soil aggregate size distribution and increases soil CEC in the Aw layer.

3. Mixing of the Aw and B layers affects soil physicochemical properties and alters SOC contents throughout the soil profile and the benefits for grain productivity could persist for two years.

Furthermore, according to the results of the present study, the maize straw amendment into the Aw layer when mixing the Aw and B layers further improved Albic soil physicochemical properties and crop yield, and the effects of the FMSP treatment were superior to those of the other treatments. Further studies should be conducted to improve Albic soils and refine the strategies of mixing of the Aw and B layer soils. Furthermore, the addition of organic substances to the 0–40 cm soil could improve soil conditions for crop growth.

## Supporting information

**S1 Fig. Fig 3. Effects of soil mixing on the water holding capacity of Albic soil.** (DOCX)

**S2 Fig. Fig 4. Effects of soil mixing on the soil penetration resistance of Albic soil.** (DOCX)

**S3 Fig. Fig 5. Effects of soil mixing on the water-stable aggregate of Albic soil.** (DOCX)

**S4 Fig. Fig 6. Effects of soil mixing on the MWD of Albic soil.** (DOC)

## Acknowledgments

The authors are grateful to the members of Jiamusi Branch of Heilongjiang Academy of Agricultural Sciences.

## Author Contributions

**Conceptualization:** Chunfeng Zhang.

**Formal analysis:** Xiaohe Yang.

**Visualization:** Yanyu Han.

**Writing – original draft:** Qingying Meng, Hongtao Zou, Chunfeng Zhang, Baoguo Zhu, Nannan Wang.

**Writing – review & editing:** Zhijia Gai.

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
