## [Decision Letter · Decision Letter 0]

15 Jul 2020

PONE-D-20-19045

Soil mixing with organic matter amendment improves Albic soil physicochemical properties and crop yield in Heilongjiang province, China

PLOS ONE

Dear Dr. Meng,

Thank you for submitting your manuscript to PLOS ONE. After careful consideration, we feel that it has merit but does not fully meet PLOS ONE’s publication criteria as it currently stands. Therefore, we invite you to submit a revised version of the manuscript that addresses the points raised during the review process.

I have to note that one of the reviewers rejected the paper. Thus, I expect a substantial revision of the manuscript since it is going to be subjected to a second review round with different reviewers. This does not guarantee that the paper will be finaly accepted.

We look forward to receiving your revised manuscript.

Kind regards,

Vassilis G. Aschonitis

Academic Editor

PLOS ONE

Journal Requirements:

Reviewers' comments:

Reviewer's Responses to Questions

**Comments to the Author**

1. Is the manuscript technically sound, and do the data support the conclusions?

Reviewer #1: Partly

Reviewer #2: Yes

2. Has the statistical analysis been performed appropriately and rigorously? 

Reviewer #1: No

Reviewer #2: Yes

3. Have the authors made all data underlying the findings in their manuscript fully available?

Reviewer #1: Yes

Reviewer #2: No

4. Is the manuscript presented in an intelligible fashion and written in standard English?

Reviewer #1: Yes

Reviewer #2: Yes

5. Review Comments to the Author

Reviewer #1: Review PONE -D-20-19045

The investigators evaluated the effects of various methods of soil subsoiling and mixing on the physical and yield characteristics of an albic soil in China. The methods show some variability from other attempts to ameliorate subsoils with poor structure and overall the mew methods were significantly better at altering subsoil structure that a standard technique used as a control. Nevertheless, the observations are not particularly original and there are issues with the description of methodology, reporting of the data, and subsequent discussion that should be addressed.

The authors use very generic terms for the soil classification they are using and to make their work more broadly available they should report their soil type in comparison to some standard method of soil classification use internationally such as the FAO. Two of the treatments reflect mixing management that incorporates straw or differential mixing. But the sampling protocol is vague as to where the composite samples came from and how the differential mixing was adequately and systematically compared.

Table 4 reports grain yields and various other agronomic properties, but they differ from year to year because of crop-to-crop variability. However, no explanation is provided for why the agronomic characteristics make sense to measure, and it is not clear to what the final column (‘increasing’) refers. Presumably it is yield differential relative to the subsoiled control.

The discussion is largely repetitive of the results and merely indicates that because of mixing some change was observed, but doesn’t provide and more substantial discussion of why or how that change occurred. It also describes various changes in agronomic practices based on the results that are not supported by research that has been conducted in the study. The long term consequences of the practices proposed are not discussed in any way.

Specific Comments

The manuscript is well-written overall and easy to follow with only the occasional grammatical quirk (e.g. line 11). The authors use far too many significant figures in the presentation of their results and often describe effects in terms of statistical significance when their data show none are present.

l. 105 This statement in inexplicable.

It should be more clearly specified when in Fall 2014 the tillage occurred and be explicit that only one tillage was performed and followed the subsequent years.

l. 126 ff Better explain what you mean by undisturbed soil samples. Presumably this means samples that were not homogenized.

l. 129 In FSMP and TSIMP describe how the sampling was done to uniformly reflect the straw addition and alternate mixing. With alternate mixing, was it alternate within a year or between years.

While it is useful to have a device that partitions soil into three phases, the variability of rain suggests that the water content will be transient. So a better reflection of the phase changes would simply be to assess total porosity, which could then be better compared to changes in bulk density. To get at the significance of water it would have been better to address pore size distribution and water potential as a function of treatment.

l. 158-159 This is repetitive of the data presentation in the table and should be revised to provide something new. Considering that the B horizon is profoundly disturbed in these treatments, it is remarkable how consistent the bulk density remain. Presumably this means the bulk density samples came from beneath where mixing occurred. In any event, it is not discussed.

l. 164-172 A very detailed description best reflected by simply reporting total porosity and shows that treatment increases total porosity relative to the control.

l. 179 This must refer to both Figures 4 and 5. For this section do a better job of defining your years to distinguish your data.

l. 203-204 Present your data chronologically from earliest to latest. Presumably this is one of the points about the duration of treatment effect you are trying to make. Although, given that the crop and drop rooting depth was different, without a true control it is impossible to tease out from the data the extent to which crop influenced the physical transformations that occurred.

Table 2 indicates that CEC varies as part of mixing, but BD does not. Discuss why.

l. 240-244 Greater in relation to what? On what basis if you have multiple measures of productivity?

l. 250 This should have the appropriate number associate with the reference [21].

l. 307 The soil with which you are dealing is only marginally acidic and no treatment indicates liming was used. So, although this is obviously sound agronomic practice when disturbing a soil, the data do not provide a rationale for why it should be done.

Reviewer #2: Writers should proofread document further. Some examples:

Lines 30 and 32 repeat the "first providing a Chinese name for Albic soil" point.

Unsure if this was intentional, but Line 38 and 39 mention the Ap layer in the same sentence.

Complete data set not attached unless I missed it. Requires attachment as per PLOS ONE policy.

6. PLOS authors have the option to publish the peer review history of their article (what does this mean?). If published, this will include your full peer review and any attached files.

Reviewer #1: No

Reviewer #2: No

---

## [Author Response · Author response to Decision Letter 0]

5 Aug 2020

Response to reviewers

Dear Editor: 

We thank you and the reviewers for your thoughtful suggestions and insights. The manuscript has benefited from these insightful suggestions. I look forward to working with you and the reviewers to move this manuscript closer to publication in the PLOS ONE.

The manuscript has been rechecked, and the necessary changes have been made in accordance with the reviewers’ suggestions. The responses to all comments have been marked in blue in the revised manuscript.

Thank you for your consideration. I look forward to hearing from you.

Sincerely,

Qingying Meng

Response to reviewer#1:

The manuscript is well-written overall and easy to follow with only the occasional grammatical quirk (e.g. line 11). The authors use far too many significant figures in the presentation of their results and often describe effects in terms of statistical significance when their data show none are present.

Response:

Thanks to reviewer. We have modified the grammar about line 11. we have reduced far too many significant figures in the presentation of our results.

l. 105 This statement in inexplicable.

It should be more clearly specified when in Fall 2014 the tillage occurred and be explicit that only one tillage was performed and followed the subsequent years.

Response:

Thanks to reviewer. We have added the sentence “No tillage was performed in 2015-2016 years of this experiment.”

l. 126 ff Better explain what you mean by undisturbed soil samples. Presumably this means samples that were not homogenized.

Response:

Thanks to reviewer. “Undisturbed soil samples were collected from the Ap and Aw layers using a wide-blade knife at five points and placed in sampling boxes and these samples were mixed for later soil aggregate measurements.”

l. 129 In FSMP and TSIMP describe how the sampling was done to uniformly reflect the straw addition and alternate mixing. With alternate mixing, was it alternate within a year or between years.

While it is useful to have a device that partitions soil into three phases, the variability of rain suggests that the water content will be transient. So a better reflection of the phase changes would simply be to assess total porosity, which could then be better compared to changes in bulk density. To get at the significance of water it would have been better to address pore size distribution and water potential as a function of treatment.

Response:

Thanks to reviewer. In this study, The experimental field was established in Autumn 2014 by four tillage systems, no tillage in 2015 and 2016. In TSIMP, soil is mixed across a 92-cm-wide site with alternating mixed and non-mixed areas, with a non-mixed 62-cm buffer between the intervals. The soil samples were collected using S-type methods at five points and make sure the five points were distributed according to the proportion of the mixed and non-mixed in TSIMP.

We have replaced the three phases of soil data with soil pore characteristics (total porosity, capillary porosity and air-porosity ) and water holding capacity in the manuscript. 

l. 158-159 This is repetitive of the data presentation in the table and should be revised to provide something new. Considering that the B horizon is profoundly disturbed in these treatments, it is remarkable how consistent the bulk density remain. Presumably this means the bulk density samples came from beneath where mixing occurred. In any event, it is not discussed.

Response:

Thanks to reviewer. We have deleted the repetitive of the data in the table and provided something in the discussion. The structure of the Aw layer soil, where clay fills the pore spaces of frame structures produced using silt particles, leads to a high bulk density. When mixing the B layer into the Aw layer, the bulk density of Aw layer was decreased. Mixing the layers changed the bulk density of B layer , but the results were similar to the before mixing.

l. 164-172 A very detailed description best reflected by simply reporting total porosity and shows that treatment increases total porosity relative to the control.

Response:

Thanks to reviewer. The total porosity, capillary porosity and air-porosity were showed in the manuscript. 

l. 179 This must refer to both Figures 4 and 5. For this section do a better job of defining your years to distinguish your data.

Response:

Thanks to reviewer. In this section Figure 5 was modified Figure 4.

l. 203-204 Present your data chronologically from earliest to latest. Presumably this is one of the points about the duration of treatment effect you are trying to make. Although, given that the crop and drop rooting depth was different, without a true control it is impossible to tease out from the data the extent to which crop influenced the physical transformations that occurred.

Table 2 indicates that CEC varies as part of mixing, but BD does not. Discuss why.

Response:

Thanks to reviewer. Yes, the data was in chronological order, and the years were modified. The soil CEC of the Aw layer was 60% lower than that of the B layer, suggesting that the fertilizer retention performance of the Aw soils was poor (Murai 1987). The CEC of the mixed soils increased following the mixing of the Aw and B layers. When mixing the B layer into the Aw layer, the bulk density of Aw layer was decreased. Mixing the layers changed the bulk density of B layer, but the results were similar to the before mixing.

l. 240-244 Greater in relation to what? On what basis if you have multiple measures of productivity?

Response:

Thanks to reviewer. We have deleted the sentence, and the final column (increasing) in table 5.

l. 250 This should have the appropriate number associate with the reference [21].

Response:

Thanks to reviewer. We have added the number associate with the reference.

l. 307 The soil with which you are dealing is only marginally acidic and no treatment indicates liming was used. So, although this is obviously sound agronomic practice when disturbing a soil, the data do not provide a rationale for why it should be done.

Response: Thanks to reviewer. In our study, the soil is only marginally acidic, so this part was deleted.

Reviewer #2: Writers should proofread document further. Some examples:

Lines 30 and 32 repeat the "first providing a Chinese name for Albic soil" point.

Response:

Thanks to reviewer. The manuscript was proofread.

Unsure if this was intentional, but Line 38 and 39 mention the Ap layer in the same sentence.

Response:

Thanks to reviewer. We have corrected this section.

Complete data set not attached unless I missed it. Requires attachment as per PLOS ONE policy.

Response:

Thanks to reviewer. We have submitted additional supporting information when I submitted the manuscript.

---

## [Decision Letter · Decision Letter 1]

14 Sep 2020

Soil mixing with organic matter amendment improves Albic soil physicochemical properties and crop yield in Heilongjiang province, China

PONE-D-20-19045R1

Dear Dr. Meng,

We’re pleased to inform you that your manuscript has been judged scientifically suitable for publication and will be formally accepted for publication once it meets all outstanding technical requirements.

Kind regards,

Vassilis G. Aschonitis

Academic Editor

PLOS ONE

Additional Editor Comments (optional):

Reviewers' comments:

Reviewer's Responses to Questions

**Comments to the Author**

1. If the authors have adequately addressed your comments raised in a previous round of review and you feel that this manuscript is now acceptable for publication, you may indicate that here to bypass the “Comments to the Author” section, enter your conflict of interest statement in the “Confidential to Editor” section, and submit your "Accept" recommendation.

Reviewer #2: (No Response)

2. Is the manuscript technically sound, and do the data support the conclusions?

Reviewer #2: Yes

3. Has the statistical analysis been performed appropriately and rigorously? 

Reviewer #2: Yes

4. Have the authors made all data underlying the findings in their manuscript fully available?

Reviewer #2: No

5. Is the manuscript presented in an intelligible fashion and written in standard English?

Reviewer #2: Yes

6. Review Comments to the Author

Reviewer #2: -

7. PLOS authors have the option to publish the peer review history of their article (what does this mean?). If published, this will include your full peer review and any attached files.

Reviewer #2: No

---

## [Editor Report · Acceptance letter]

2 Oct 2020

PONE-D-20-19045R1 

Soil mixing with organic matter amendment improves Albic soil physicochemical properties and crop yield in Heilongjiang province, China 

Dear Dr. Meng:

I'm pleased to inform you that your manuscript has been deemed suitable for publication in PLOS ONE. Congratulations! Your manuscript is now with our production department. 

Kind regards, 

on behalf of

Dr. Vassilis G. Aschonitis 

Academic Editor

PLOS ONE